

# Weak Southern Hemispheric monsoons during the Last Interglacial period

Nicholas K. H. Yeung[1,2], Laurie Menviel[1], Katrin J. Meissner[1,2], Andréa S. Taschetto[1,2], Tilo Ziehn[3], and Matthew Chamberlain[4]

[1]Climate Change Research Centre, University of New South Wales, Sydney NSW, Australia
[2]ARC Centre of Excellence for Climate Extremes, University of New South Wales, Sydney NSW, Australia
[3]CSIRO Oceans and Atmosphere, Aspendale VIC, Australia
[4]CSIRO Oceans and Atmosphere, Hobart TAS, Australia

**Correspondence:** Nicholas K. H. Yeung (nicholas.yeung@unsw.edu.au)

**Abstract.** Due to different orbital configurations, high northern latitude boreal summer insolation was higher during the Last Interglacial period (LIG; 129–116 thousand years before present, ka) than during the preindustrial period (PI), while high southern latitude austral summer insolation was lower. The climatic response to these changes is studied here with focus on the southern hemispheric monsoons, by performing an equilibrium experiment of the LIG at 127 ka with the Australian Earth System Model, ACCESS-ESM1.5, as part of the Paleoclimate Model Intercomparison Project 4 (PMIP4). In our simulation, mean

surface air temperature increases by 6.5 °C over land during boreal summer between 40° N and 60° N in the LIG compared to PI, leading to a northward shift of the Inter-Tropical Convergence Zone (ITCZ) and a strengthening of the North African and Indian monsoons. Despite 0.4 °C cooler conditions in austral summer in the Southern Hemisphere (0–90° S), annual mean air temperatures are 1.2 °C higher at southern mid-to-high latitudes (40° S–80° S). These differences in temperature are coincident

with a large-scale reorganisation of the atmospheric circulation. The ITCZ shifts southward in the Atlantic and Indian sectors during the LIG austral summer compared to PI, leading to increased precipitation over the southern tropical oceans. However, the decline in Southern Hemisphere insolation during austral summer induces a significant cooling over land, which in turn weakens the land-sea temperature contrast, leading to an overall reduction (−20 %) in monsoonal precipitation over the Southern Hemisphere's continental regions. The intensity and areal extent of the Australian, South American and South African

monsoons are consistently reduced. This is associated with greater pressure and subsidence over land due to a strengthening of the southern hemispheric Hadley cell during austral summer.

## 1  Introduction

Antarctic ice cores suggest that the Last Interglacial period (LIG; ∼129–116 thousand years before present (ka)), also known as Marine Isotope Stage 5e or the Eemian, was most likely the warmest interglacial of the last 800 ka (Masson-Delmotte et al.,

2013). Paleoproxy records suggest that annual mean sea surface temperatures (SSTs) were about ∼0.5 °C (Hoffman et al., 2017) to ∼1.1 °C (Capron et al., 2017) above preindustrial (PI) values at the LIG in the North Atlantic. The summer warming at high northern latitudes on land was particularly pronounced with estimated temperatures 4 to 5 °C above PI (CAPE-Last





Interglacial Project Members, 2006) and a 3 to 11 °C warming over Greenland (NEEM community members et al., 2013;
Landais et al., 2016). It has also been suggested that LIG Southern Ocean SSTs were ∼1.8 °C higher than PI (Capron et al.,
25  2017).

   Global mean sea-level was ∼6 to 9 m higher during the LIG than the PI (Kopp et al., 2009; Dutton and Lambeck, 2012),
with Greenland ice sheets contributing 0.6 to 3.5 m (Dutton et al., 2015), and an Antarctic ice-sheet contribution likely greater
than 6 m (Kopp et al., 2009; Rohling et al., 2019). Despite its importance, there are still a lot of uncertainties associated with
the processes leading to ice-mass loss at the LIG. Even though paleoproxy records provide constraints on the LIG climate, low
spatial and temporal resolution and uncertainties in transfer functions add uncertainties to the climate response to LIG boundary
conditions. Numerical simulations of the LIG can thus improve our understanding of the climate processes and feedbacks at
play.

   The LIG equilibrium simulation (lig127k) is one of the highest-priority experiments of the Paleoclimate Modeling Inter-
comparison Project 4 (PMIP4) designated simulations in the Coupled Model Intercomparison Project (CMIP6) (Otto-Bliesner
et al., 2017). It primarily aims to examine the climate response due to changes in orbital configuration at a time when atmo-
spheric greenhouse gas concentrations and continental configurations were similar to PI (Table 1). At the LIG, the Earth's orbit
had a larger eccentricity, with the timing of perihelion closer to the boreal summer solstice (Berger, 1978). Together with a
greater axial tilt of the Earth, it led to higher insolation north of 40° S between April and September, with a maximum anomaly
of ∼70 W m$^{-2}$ at high northern latitudes in June. Insolation was generally lower south of 60° N between October and March
and particularly in the Southern Hemisphere (SH) in December and January, with insolation anomalies reaching −45 W m$^{-2}$
compared to PI (Fig. 1).

   A recent study presented the large-scale features of the PMIP4-CMIP6 lig127k experiment as deduced from 17 participat-
ing climate models (Otto-Bliesner et al., 2020). Compared to PI, strong warming is shown over Northern Hemisphere (NH)
continents during June, July, and August (JJA), while a cooling is simulated in December, January, and February (DJF), due
to the seasonal character of insolation anomalies (Otto-Bliesner et al., 2020). This leads to a substantial reduction in the boreal
summer Arctic sea-ice extent, while there is little change in maximum sea-ice area during winter (Kageyama et al., 2020).
This multi-model study is in broad agreement with available paleoproxy records showing that regions south of 78° N in the
Atlantic and Nordic Seas were seasonally ice-free, though model-data comparison north of 78° N is difficult due to ambiguous
interpretations of proxy data.

The PMIP4 lig127k ensemble mean shows that summer monsoonal precipitation is enhanced over northern Africa (and
extends into Saudi Arabia), India and southeast Asia, and northwestern Mexico; while in the SH, summer monsoonal precip-
itation decreases over Australia, southern Africa and South America (Otto-Bliesner et al., 2020). Although the model spread
for precipitation is large, the models generally agree on the sign of change in area-averaged monsoonal precipitation, except
for the South Asian and Australian monsoons. While studies based on proxy records consistently demonstrate a strengthening
of the African and Indian monsoons (e.g. Magiera et al., 2019; Orland et al., 2019; Rohling et al., 2002), the picture in the SH
is less clear. A compilation of LIG precipitation proxy records with near-global coverage and a multi-model-data comparison
were presented in Scussolini et al. (2019). It includes 138 sites based on a range of proxies. Most of the data agrees that there





is generally higher northern hemispheric annual precipitation during the LIG relative to the PI, with the exception of a small number of individual proxy sites. In the SH, the proxy signal is less consistent and spatially heterogeneous, with only partial
model-data agreement.

In this study we present the large-scale climatic features of the LIG equilibrium experiment (lig127k) as simulated by the ACCESS-ESM1.5 model, compared to the preindustrial experiment (Ziehn et al., 2020), and to available paleoproxy records. We also explore the changes in austral summer precipitation in the SH.

## 2  Model description and experimental design

### 2.1  ACCESS-ESM1.5 model description

ACCESS-ESM1.5 (Ziehn et al., 2020) is an updated version of ACCESS-ESM1 (Law et al., 2017). The differences between ACCESS-ESM1.5 and ACCESS-ESM1 are relatively small, with the majority of the changes concerning the land surface and ocean model.

The atmospheric component of the ACCESS-ESM1.5 is the UK Met Office Unified Model version 7.3 (UM7.3; Martin
et al., 2010; The HadGEM2 Development Team et al., 2011), but with the Community Atmosphere Biosphere Land Exchange model (CABLE2.4; Kowalczyk et al., 2013) as land surface model. The ocean component is the NOAA/ GFDL Modular Ocean Model version 5 (MOM5; Griffies, 2014) with the same configuration as the ocean model component of ACCESS1.0 and ACCESS1.3 (Bi et al., 2012). Sea ice is simulated using the LANL CICE4.1 model (Hunke and Lipscomb, 2010), which has the same horizontal grid as the ocean with 5 thickness classes. Coupling of the ocean and sea-ice to the atmosphere is
achieved through the OASIS-MCT coupler (Valcke, 2013). The physical climate model configuration used here is similar to the ACCESS1.3 model, which contributed to the Coupled Model Intercomparison Project Phase 5 (CMIP5) (Bi et al., 2012). The spatial resolutions of model components are listed in Table 1.

The carbon cycle is included in ACCESS through the Nutrient, Phytoplankton, Zooplankton and Detritus (NPZD) model WOMBAT (World Ocean Model of Biogeochemistry and Trophic dynamics; Oke et al., 2013), and through the land surface
model CABLE and its biogeochemistry module CASA-CNP (Wang et al., 2010), with CASA-CNP being run with nitrogen and phosphorus limitation.

In the CABLE configuration applied here, there are a total of 13 plant functional types (PFTs), including 10 vegetated types and 3 non-vegetated types (lake, land-ice, bare ground). For land-ice, CABLE does not allow fractional amount such that relevant grid cells must be all permanent ice, effectively limiting these cells to Greenland and Antarctica. CABLE calculates
gross primary production (GPP) and leaf respiration at every time step using a two-leaf canopy scheme (Wang and Leuning, 1998) as a function of the leaf area index (LAI). LAI is calculated prognostically based on the size of the leaf carbon pool and the specific leaf area. Here the PFT is fixed such that vegetation is static, but LAI is interactive.

Biases in the ACCESS-ESM1.5 are discussed in Ziehn et al. (2020). In boreal summer, India and North America show signs of a warm bias, with a dry bias over India. In austral summer, there are warm biases over the equatorial regions in South





America and Africa, with both wet and dry biases over land. By comparison, Australia's biases in temperature and precipitation
    are very small.

## 2.2   Experimental design

The Last Interglacial is one of the two interglacial periods included in PMIP4. The equilibrium experiment of the Last Inter-
glacial, denoted lig127k, is classified as a Tier 1 PMIP4-CMIP6 experiment.

The initial conditions of the lig127k experiment were derived from a preindustrial simulation (1850 CE, piControl) (Ziehn
    et al., 2020), which follows the CMIP6 protocol (Eyring et al., 2016). piControl was integrated for 1000 years, and the average
    of the last 100 years serves as a reference to which the lig127k experiment is compared.

The lig127k experiment follows the lig127k protocol (Table 1, Otto-Bliesner et al. (2017)), with specific forcing values
described in Table 1. As the solar constant in the piControl experiment follows CMIP5-PMIP3 guidelines (1365.65 W m$^{-2}$),
it has a slightly different value to CMIP6 protocol (1360.75 W m$^{-2}$). The solar constant in lig127k is set equal to the one in
    piControl to allow a direct comparison between them. The experiment is integrated for 650 years, and we are presenting the
    last 200 years of that run. During the last 100 years, changes in globally averaged SST are +0.11 °C, changes in deep ocean
    temperature are +0.043 °C and changes in salinity in the Southern Ocean are less than 0.005 psu, which suggests that our
    experiment has equilibrated sufficiently.

Since the orbital parameters of LIG and PI are different (Table 1), a fixed-angular definition of months is required to achieve
    a valid comparison between lig127k, piControl, and proxy data. The length of each month should be defined by a fixed number
    of degrees of the Earth's orbit, as opposed to number of days. Therefore, the outputs of lig127k are adjusted following Bartlein
    and Shafer (2019). It is essential to consider this paleo-calendar effect for a correct interpretation of results.

## 3   Results

As the maximum insolation anomalies between the LIG and PI occur in June (+70 W m$^{-2}$ at 80° N) and December (−45 W
    m$^{-2}$ at 40° S) (Fig. 1), we here focus on climatic changes occurring in JJA and DJF.

### 3.1   Changes in surface temperature and sea-ice

Due to the large insolation anomalies in the NH during boreal summer (Fig. 1), simulated mean JJA surface air temperatures
are 2.3 °C higher in the NH at the LIG compared to PI (Fig. 2b), in agreement with terrestrial proxy reconstructions from the
region (Axford et al., 2011; Francis et al., 2006; Fréchette et al., 2006; McFarlin et al., 2018; Melles et al., 2012; Plikk et al.,
    2019; Salonen et al., 2018). The simulated boreal summer warming is maximum between 40° N and 60° N, averaging +6.5
    °C over land, in line with the PMIP4 lig127k multi-model mean (Otto-Bliesner et al., 2020). Similarly, compared to the PI
    simulation, boreal summer SSTs are ∼2.5 °C higher in the North Pacific, the North Atlantic and the Nordic Seas, and up to
    4 °C higher in the Labrador Sea (Fig. 2d). This high latitude warming is associated with a reduction in boreal summer Arctic
sea-ice cover (Fig. 2d), with a maximum of 67 % reduction in sea-ice cover in September (Figs. S1, S2). North of 40° N in





the North Atlantic and Norwegian Sea, 16 paleoproxy records suggest higher SSTs during boreal summer at the LIG and 9 suggest lower SSTs, with a range of $-8.7$°C to $+5.7$ °C, and a median of $+1.5$ °C (Fig. 2d). Out of these 25 records, simulated SSTs agree with the anomaly sign of 16 of the paleoproxy records. The main regions of model-data disagreement, where proxy records suggest a cooling whereas the model suggests a warming, are in the Norwegian Sea and off the Iberian margin.

Due to the strong sea-ice melting in boreal summer, the simulated boreal winter Arctic sea-ice cover remains 8 % smaller at the LIG compared to PI in spite of a higher rate of boreal autumn sea-ice formation (Fig. 2c, S1). In addition, enhanced deep water formation is simulated in the Labrador Sea during the LIG (not shown), inducing a $\sim$2.5 °C increase in winter SSTs in that region (Fig. 2c). However, the simulated SST increase in the Labrador Sea during JJA is in contrast with some of the SST paleoproxy records which show cooling instead (Fig. 2d; Capron et al., 2014, 2017). Furthermore, there is no proxy evidence

for a deep water formation site in the Labrador Sea during the LIG (Hillaire-Marcel et al., 2001). The simulated deep water formation site might therefore be in the wrong location, thus explaining the discrepancy.

The simulated strength of the Atlantic Meridional Overturning Circulation (AMOC), as represented by the maximum meridional stream function at 26 °N in the Atlantic basin, is stronger at the LIG (21.8 Sv) than during PI (18.3 Sv), with enhanced deep-ocean convection in the Labrador Sea. The AMOC strengthening and reduced Arctic sea-ice cover lead to higher winter

SST in the northern North Atlantic by  3°C and air surface temperatures over the ocean at high northern latitudes (+3.3 °C in 60–90° N) (Fig. 2a, c). As a result, the annual mean northern hemispheric (0–90° N) surface air temperature is 1.4 °C higher at the LIG compared to PI (Fig. 3b).

Due to the lower magnitude of insolation in austral summer, simulated DJF air temperatures are lower at mid and low latitudes at the LIG compared to PI. This cooling is enhanced over land, where the anomalies can be as low as $-5.4$ °C in

India (Fig. 2a). SSTs between 40° N and 40° S also drop by 0.5 °C on average during DJF, in good agreement with most proxy records in mid-latitudes. The largest SST drop at low-latitudes is simulated in the Bay of Bengal, with an anomaly of $-3$ °C (Fig. 2c). A $\sim$2 °C SST decrease is also simulated in the western equatorial Pacific and in the equatorial Atlantic. However, despite lower austral summer insolation at high southern latitudes, warmer conditions are simulated in DJF at the LIG compared to PI in the Southern Ocean, associated with a large decrease in sea-ice extent ($-44$ %) (Figs. 2a, 2c and S1).

Compared to PI, the insolation reaching 60° S at the LIG is only 5 to 15 W m$^{-2}$ larger between mid April and mid September (Fig. 1). However, this relatively small positive insolation anomaly reduces the growth of Antarctic sea-ice between April and September, with the simulated maximum Antarctic sea-ice extent in September being 32 % smaller during the LIG than during PI (Fig. S1). Despite lower insolation, Antarctic sea-ice extent reaches a minimum in February that is 48 % smaller at the LIG than at PI. This prominent reduction in Antarctic sea-ice extent is accompanied by a marked temperature increase in the

Southern Ocean all year round (Fig. 2). Simulated SSTs are 2 to 3 °C higher in the Southern Ocean during austral summer (Fig. 2d), while annual mean air temperatures over Antarctica increase close to the coast by $\sim$4 °C (Fig. 3b). The simulated Southern Ocean warming between 50° S and 60° S is in agreement with SST proxy records, however it is underestimated between $\sim$40° S and 50° S (Capron et al., 2017) (Fig. 2c). While the simulated warming is mostly confined to the south of 45° S, proxy records suggest the warming could have reached further north, particularly in the Atlantic Ocean. In short, annual

mean surface air temperature is 0.5 °C warmer in the SH, and it is 1.3 °C warmer averaged south of 50° S (Fig. 3b).



## 3.2 Precipitation change

Simulated annual mean precipitation anomalies are shown in Figure 3a, and compared to a recent compilation of LIG precipitation reconstructions based on a range of proxy records (including pollen, speleothems, landscape features, loess and sediment composition) (Scussolini et al., 2019). In the NH the model is in agreement with 64 out of 109 proxy records (59 %) (where

the model and proxy data show the same sign of change, or where the change in the simulations is <100 mm yr$^{-1}$ and the corresponding proxy suggests no change). Although the agreement is not compellingly strong, it is worth pointing out that the majority of disagreements arise in central Europe where reconstructions are abundant and mostly display wetter conditions, whereas the simulation shows a slight decrease in precipitation. In contrast, there is a good model-data agreement in northern Africa, the Middle East, Asia and North America.

As seen in Figure 3a, the largest changes in annual mean precipitation are simulated in the tropics where both the rainfall mean and variability are higher. Annual precipitation increases over land in the northern tropics, particularly over North Africa (the Sahel, >+300 %), South Asia (India, +100 %), and Central America (Mexico, +40 %). Coinciding with increasing precipitation, the significant cooling ($\sim -3$ °C; Fig. 3b) over the Sahel and India is associated with stronger evaporation over land during boreal summer (not shown). All ocean regions in the northern tropics are simulated to be wetter, except for the South

China Sea and Philippine Sea. In general there is good agreement with proxy records in the NH.

In contrast, the model generally simulates drier conditions over the southern tropics during the LIG (Fig. 3a), particularly over South America (−20 %), South Africa (−40 %) and northern Australia (−40 %). This decrease in precipitation mostly occurs in DJF, during the Southern Hemispheric monsoon season (Fig. 4a). In addition, this precipitation reduction is consistent with an overall increase in mean sea-level pressure over the SH land (Fig. 4b). Therefore, the simulation suggests a major

change in precipitation in the tropics, associated with a shift in the Inter-Tropical Convergence Zone (ITCZ) and a weakening of the convergence zones in the SH, which will be discussed in the next section.

### 3.2.1 ITCZ changes

The ITCZ position corresponds to the latitudinal band of maximum precipitation. Previous studies have demonstrated that the position of the ITCZ aligns with the zero energy flux equator, i.e. where the atmospheric meridional energy flux divergence

vanishes (Schneider et al., 2014; Ceppi et al., 2013). In the present climate, this occurs north of the equator, thus placing the ITCZ in the NH. This is essentially due to an interhemispheric energy imbalance, caused primarily by the northward transport by the Atlantic Meridional Overturning Circulation (AMOC). The net northward oceanic heat transport to the NH is compensated by a southward atmospheric heat transport via a northward-displaced Hadley cell and the positioning of the ITCZ north of the Equator. Therefore, one would expect that an interhemispheric temperature gradient would drive a cross-equatorial

atmospheric energy flux, which in turn determines the position of the ITCZ towards the warmer hemisphere (e.g., Schneider et al., 2014).

As seen in Figure 5h, zonally averaged precipitation in JJA displays a slight northward shift of the ITCZ at the LIG compared to PI. Although this northward ITCZ shift is seen in all ocean basins, there are differences in the magnitude of the peak





precipitation. The peak is 9 % stronger in the Pacific sector (130° E to 70° W), but 39 % weaker in the Atlantic sector. In the
Indian sector (0 to 130° E), even though the ITCZ location is less defined, precipitation is higher between 5° S and 20° N at the
LIG. In short, the ITCZ shifts northward in the Pacific and Atlantic sectors in JJA compared to PI. This consistent northward
shift during JJA can be explained by the higher NH summer temperatures, which accentuate the interhemispheric temperature
gradient.

The globally averaged precipitation in DJF suggests a contraction of the ITCZ, with an increase in strength (by 10 %) at
5° S, while precipitation at 5° N decreases by 23 % (Fig. 5d). However, the DJF precipitation response across basins varies
significantly. In the Indian sector, there is a southward shift (∼3°) and slight strengthening of the ITCZ. In contrast, in the
Pacific sector, which displays a double ITCZ in DJF in both PI and LIG, a weakening and northward shift (∼1° to 4°) of the
precipitation peaks at 10° S and 7° N are simulated. Similarly, in the Atlantic sector, a ∼25 % weakening of DJF precipitation
at 5° N during the LIG is simulated, while there is a strengthening of the ITCZ peaks at 5° S (Figs. 4a, 5c). The simulation
thus displays a southward ITCZ shift over the Atlantic and Indian Oceans, but a northward shift in the Pacific Ocean. These
differences in longitudinal responses are associated with changes in large-scale atmospheric circulation. Since low latitude
precipitation is associated with monsoon systems, we will now look into more detail at the precipitation changes occurring in
each of the monsoon regions.

### 3.2.2 Precipitation changes in monsoon regions

Monsoon domains are defined as regions in which the monsoon season precipitation is greater than 2.5 mm day$^{-1}$ compared
to dry season (Wang et al., 2011). In the NH, the May-to-September (general monsoon season) precipitation should therefore
be 2.5 mm day$^{-1}$ higher than the November-to-March (dry season) precipitation to qualify as a monsoon region (Wang et al.,
2011). The reverse holds for the SH: precipitation during November to March is required to be 2.5 mm day$^{-1}$ higher than
May to September. As shown in Figure 6a, the model simulates a general extension of the monsoon domains in the northern
tropics, also associated with increased precipitation rates, whereas there is a contraction of the monsoon domains and reduced
precipitation in the southern tropics.

The North African monsoon domain expands significantly into the Sahara region. This expansion is associated with enhanced
southwesterly winds. Indian monsoon also strengthens with monsoon domain covering most of India and stronger onshore
winds from the Arabian Sea and convergence inland (Fig. 6a). On the other hand, a contraction of the monsoon domain over
the Philippine Sea and South China Sea is simulated and attributed to a northeastern wind anomaly associated with trade winds
strengthening in JJA.

As evident from Figure 6b, there is a marked difference in DJF precipitation anomalies over the SH land and ocean regions
during LIG relative to PI. To investigate this difference further, we separate the simulated tropical precipitation anomalies
over land and ocean (Fig. 7). Precipitation over land is generally higher in the northern tropics, with an increase of over 60
% in total precipitated water for the North African monsoon (NAF) (Fig. 7). The South Asian Monsoon (SA) also displays a
large increase in precipitated water (>60 %), mainly due to a large increase in areal-extent (+>55 %), as the increase in area-
averaged precipitation is small (∼5 %). The North American monsoon (NAM) is slightly weakened, with a slight decrease in





area-averaged precipitation rate (−1.4 %), areal extent (−2.2 %) and total precipitated water (−3.6 %). All monsoon domains in the NH experience an increase in air surface temperature during monsoon seasons.

The contrary is simulated in the SH, with higher precipitation simulated over the ocean and lower precipitation, by at least 10 %, over land in the SH monsoon regions at the LIG compared to PI (Figs. 6b and 7). In the Indo-Australian region, there is a northward shift and weakening of the South Pacific Convergence Zone (SPCZ). This leads to a precipitation increase over the western equatorial Pacific. In contrast, the Australian monsoon (AUS) displays the greatest decline, with a decrease in total precipitated water over land by over 60 %, and a ∼50 % decrease in areal extent as the monsoon domain contracts

northward. Drier conditions ($<-1$ mm day$^{-1}$) are also simulated over Eastern Australia, even though it is not being included in the monsoon domain.

    In contrast, the Indian Ocean Convergence Zone (IOCZ) shifts southward, which leads to a ∼8 % intensification of precipitation in the Indian Ocean between 10° S and 20° S, associated with a strengthening of northwesterly surface wind north of the IOCZ (Figs. 6b and 7). This southward shift is associated with an IOCZ weakening, which induces a 35 % decrease in

total monsoonal precipitation over land in the South African monsoon region (SAF), and a 15 % decrease in the areal extent of the SAF. Central southern Africa (30° E, 15° S) displays the greatest decrease in summer precipitation. The monsoon domain contracts along the north-south direction, but expands westward close to the equator on the west coast (15° E, 5° S). This could be linked to the southward shift of the ITCZ in the southern tropical Atlantic Ocean (Fig. 6b).

    Area-averaged precipitation associated with the South American monsoon (SAM) decreases by 20 % (Fig. 7), while the

areal extent only decreases slightly as the monsoon domain remains spatially very similar to the PI (Fig. 6b). This means that the monsoon reduction is primarily due to weaker precipitation over the same region, with drier conditions more prevalent over the west coast and the southern boundary of the monsoon domain (25° S), suggesting a weakening of the South Atlantic Convergence Zone (SACZ; Carvalho et al., 2002). The Brazilian coast along 0 to 15° S shows an increase in summer precipitation (+1 to +3 mm day$^{-1}$) due to the southward shift of the ITCZ in the Atlantic Ocean, but this does not contribute much to the

general monsoon activity since LIG rainfall levels remain low.

    To a first order, the reduced precipitation over land in the southern tropics can be explained by the consistently colder conditions over land masses in SH low to mid- latitudes during austral summer (Fig. 2a). Due to reduced land-sea temperature gradients, colder conditions over land induce a weakening of the onshore winds (i.e. a weakening of the easterlies over Brazil, South Africa, and NE Australia), which tends to decreases moisture advection inland, and restricts convective activity over the

SH land.

    Due to the high heat capacity of the ocean, land masses are more sensitive to changes in insolation. As insolation is lower across most latitudes in LIG compared to PI during austral summer, the strongest anomalous cooling therefore occurs over land (Fig. 2a), which leads to positive surface pressure anomalies (Fig. 4b). In particular, the strong cooling over southwestern Australia (Fig. 2a) is associated with an anomalously high surface pressure (Fig. 4b), blocking the monsoonal inflow. The

situation is similar in India: the strongest cooling occurs over the region centered on India, which induces positive pressure anomalies over the region and the northern part of the Indian Ocean. On the other hand, negative pressure anomalies develop in the southern tropical Indian Ocean. As a result the IOCZ shifts southward.





While the reduced insolation in the SH during DJF would tend to reduce precipitation over the SH, changes in the global atmospheric circulation produce a southward shift of the ITCZ in the Indian and Atlantic Oceans (Sect. 3.2.1, Figs. 4a, 5a,

and 5c) that counteract the insolation changes and lead to higher precipitation over those tropical ocean regions (Fig. 7). In addition, the land-ocean contrast helps maintain this configuration via changes in zonal pressure gradient between land and adjacent oceans, creating local Walker-type circulation anomalies (with anomalous ascending motion over oceans and compensatory subsidence over land).

A strengthening of the Hadley circulation in the SH is simulated in DJF (Fig. 8). It contributes to the simulated drier

conditions in the subtropics at ∼30° S due to greater subsidence, as seen from the increase of surface pressures over land. The southern boundary of the Hadley cell also experiences a slight northward shift, from ∼31° S to ∼28° S, which pushes regions of high pressure in the subtropics to the north (Fig. 4b). This favours an anomalous subsidence and weaker convection over the SH convergence zones thus reducing monsoonal precipitation.

## 4 Discussion

At the LIG the insolation reaching Earth was different from PI, with higher insolation in JJA in the NH and lower insolation in DJF in the SH. In agreement with paleo-records, the ACCESS-ESM1.5 lig127k simulation presented here suggests a 2 to 4 °C SST increase in the North Atlantic as well as a 5 °C warming over land in the mid-northern latitudes in JJA. The simulated northern latitude temperature anomalies as well as JJA Arctic sea-ice cover are very close to the PMIP4 lig127k multi-model means (Kageyama et al., 2020; Otto-Bliesner et al., 2020). The simulated Arctic LIG sea-ice cover is in agreement with 16

out of 27 proxy records, and suggests that the simulated LIG Arctic sea-ice cover might be slightly overestimated (Kageyama et al., 2020).

At low to mid southern latitudes, simulated surface air temperature anomalies in the ACCESS-ESM are also very similar to the PMIP4 multi-model mean with ∼3 °C warming over land in JJA, and ∼2 °C cooling over land in DJF. However, the model simulates much warmer conditions over the Southern Ocean all year round, with values higher than the multi-model mean.

This is concurrent with a large decrease in Southern Ocean sea-ice at the LIG. While the simulated warming over the Southern Ocean throughout the year is at the higher end of the PMIP4 lig127k multi-model mean, the warming over Antarctica is at the lower end (Otto-Bliesner et al., 2020).

The simulated spatial distribution of annual precipitation anomalies (Fig. 3a) agrees well with the PMIP4 multi-model mean (Otto-Bliesner et al., 2020), with drier conditions over SH land regions including North Australia, South Africa and

South America, while wetter conditions are simulated at ∼10° S in the Atlantic Ocean, the Indian Ocean and the western equatorial Pacific Ocean. However, while our simulation suggests wetter conditions over the equatorial Pacific at 10° N due to a strengthening of the ITCZ in this basin in JJA (Fig. 5f), this does not occur in the multi-model mean.

Reduced precipitation over land and increased precipitation over the ocean in DJF in low to mid-southern latitudes is also evident in the PMIP4 multi-model mean, indicating that the ACCESS-ESM1.5 lig127k simulation provides a good representation

of the LIG features as simulated by coupled climate models. Despite the strong signal emanating from the lig127k simulations,





these precipitation anomalies are not necessarily evident from SH paleoproxy records, highlighting the need for additional hydrological records from low- and mid- southern latitudes. Furthermore, the lig127k is an equilibrium simulation, which does not take into account potential meltwater discharges from the Antarctic and Greenland ice-sheets, and their associated impact on deep water formation and therefore variability in ocean and atmospheric circulation (e.g. Hayes et al., 2014; Rohling et al.,

2019; Tzedakis et al., 2018). The time evolution across the LIG of the hydrological proxy records should thus be looked at in detail, and additional LIG experiments including North Atlantic and/or Southern Ocean meltwater pulses should be performed.

Additional studies looking into the details of the mechanisms leading to changes in Southern Hemispheric monsoon systems, and their relationship to low latitude modes of variability such as El Niño Southern Oscillation and the Indian Ocean Dipole are needed. For example, El Niño may bring sustained warming and dry conditions over eastern Australia, but wet conditions

over southeast South America. A positive Indian Ocean Dipole on the other hand could lead to increased precipitation over East Africa but droughts over southeast Australia. Under LIG conditions, these modes of variability might be altered in terms of magnitude and spatial variation.

A significant reduction in precipitation in the tropical regions of the SH would have impacted vegetation during LIG. Changes in vegetation cover are not taken into account in our study, but the LAI was interactive in the model. The simulated change in

LAI is consistent with precipitation. For example, the LAI is decreased during boreal summer and autumn at $\sim$50° N across Eurasia and North America in regions with less summer rainfall during the LIG (Fig. S3). In contrast, increased precipitation over the Sahel and surrounding regions (Fig. 6a) corresponds to elevated LAI. This affects evapotranspiration, as an increase in LAI enhances canopy evapotranspiration, while reducing soil evaporation, and ultimately also alters the hydrological cycle.

It has been previously shown that due to changes in albedo and moisture availability, changes in vegetation might amplify

the changes in precipitation (e.g. Messori et al., 2019). For example, a large increase in precipitation over the Sahel would lead to a greening of the Sahara (Hopcroft et al., 2017; Larrasoaña et al., 2013; Osborne et al., 2008; Pausata et al., 2020). The greening would tend to increase the sensible and latent heat fluxes into the atmosphere, and produce a cyclonic circulation anomaly over the Sahara, whose anomalous westerly flow would transport more moisture from the neighbouring Atlantic Ocean into the region (Messori et al., 2019; Patricola and Cook, 2007; Rachmayani et al., 2015). Additional simulations with

interactive vegetation, or with prescribed changes in vegetation cover, should thus be performed to quantify the coupled effect of precipitation and vegetation changes.

Due to the importance of the southern hemispheric monsoon systems for water availability, such reduced precipitation in the southern tropics in a past warmer world are concerning. However, since these negative precipitation anomalies mostly result from the strong cooling over land, such hydrological changes are not currently expected over the coming centuries under the

RCP/SSP (Representative Concentration Pathway / Shared Socioeconomic Pathways) scenarios where future climate changes are due to greenhouse gas forcing and not to changes in orbital parameters. CMIP6 future projections indeed generally simulate no significant changes in the Australian, South African and South American monsoons by 2100 under all SSP scenarios, while drier conditions are simulated over these regions in JJA (Cook et al., 2020). In Australia, northern rainfall in DJFMAM is more constrained in CMIP6 than in CMIP5 and it is projected to experience a $\sim$2 % increase by the year 2090 under high-emission



scenario (RCP8.5 and SSP5-85; Grose et al., 2020), which is a small change compared to the decrease in our LIG simulation (Fig. 7).

In conclusion, drier conditions are simulated over SH land during austral summer in the lig127k experiment and all southern hemispheric monsoons are found to be consistently weaker compared to PI. At the same time, precipitation over southern tropical oceans is higher, associated with a southward shift of the ITCZ in the Indian and Atlantic Oceans. The dry conditions

are caused by reduced land-sea temperature contrast due to cooling over land, and also higher mean sea level pressure attributed from greater subsidence as the southern hemispheric Hadley cell strengthens.

*Data availability.* The model results of the lig127k simulation is archived on the CMIP6 ESGF website at https://doi.org/10.22033/ESGF/CMIP6.13703

*Author contributions.* NY performed the bulk of model integration, analysis and writing. LM, KJM and AST provided support to the interpretation of results and writing. TZ and MC contributed to the model setup and troubleshooting.

*Competing interests.* The authors declare no competing interests.

*Acknowledgements.* Computational resources were provided by the NCI National Facility at the Australian National University, through awards under the Merit Allocation Scheme, the Intersect allocation scheme, and the UNSW HPC at NCI scheme. KJM, LM and AST acknowledge support from the Australian Research Council (DP180100048, FT180100606 and FT160100495). NY also acknowledges the Research Training Program provided by the Australian government, a top-up scholarship provided by the Climate Change Research Centre,

and support from the ARC Centre of Excellence for Climate Extremes.





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





**Figure 1.** Monthly insolation anomalies at LIG compared to PI.

**Figure 2.** DJF and JJA surface temperature anomalies (LIG minus PI) with proxy data overlaid (filled markers: squares for the compilation by Capron et al. (2014, 2017); dots for the compilation by Hoffman et al. (2017); triangles for Arctic terrestrial proxies (Axford et al., 2011; Francis et al., 2006; Fréchette et al., 2006; McFarlin et al., 2018; Melles et al., 2012; Plikk et al., 2019; Salonen et al., 2018)). The preindustrial reference is 1850 CE for model anomalies, 1870–1899 for Capron et al. (2014, 2017) and 1870–1889 for Hoffman et al. (2017). (a)–(b): Surface air temperatures. (c)–(d): Sea surface temperatures, with contours of sea-ice concentration at 15 % overlaid in February for DJF and August for JJA (blue: LIG; yellow: PI).







**Figure 3.** (a): Simulated annual mean precipitation anomaly (mm yr$^{-1}$) with proxy reconstructions from Scussolini et al. (2019) overlaid. Proxy reconstructions are semi-quantitative with much drier conditions shown in brown, drier in light brown, and the reverse for green. (b): Annual mean surface air temperature anomalies (°C) with proxy data (filled markers: squares for the compilation by Capron et al. (2014, 2017); triangles for Arctic terrestrial proxies (Landais et al., 2016; NEEM community members et al., 2013; Yau et al., 2016)). (c): Annual mean sea surface temperature anomalies with proxy data (filled markers: squares for the compilation by Capron et al. (2014, 2017); dots for the compilation by Hoffman et al. (2017)), and contours of annual sea-ice concentration at 15 % overlaid (blue: LIG; yellow: PI).





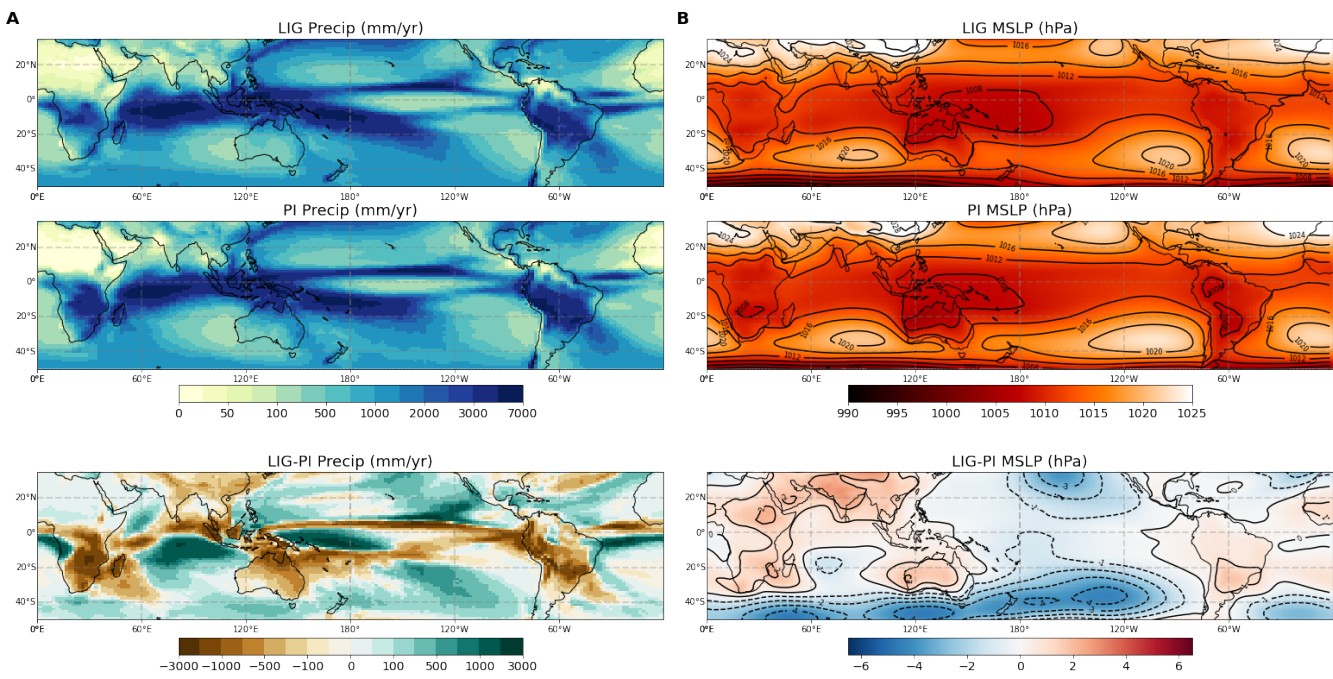

**Figure 4.** Left: Simulated DJF precipitation for LIG, PI and anomaly (LIG minus PI). Right: Simulated DJF mean sea level pressure (MSLP) for LIG, PI and anomaly (LIG minus PI).





**Figure 5.** Zonally averaged precipitation (mm day$^{-1}$). 0°–130° E covers most of Africa, Indian Ocean, and up to Western Australia; 130° E–70° W covers the Pacific Ocean; and 70° W–0° covers the central part of Atlantic Ocean. (a)–(d): DJF; (e)–(h): JJA.





**Figure 6.** Precipitation anomaly (mm day$^{-1}$) with surface wind anomaly overlaid. (a): JJA; (b): DJF. Monsoon domains defined as regions in which the monsoon season (NH: MJJAS; SH: NDJFM) precipitation is greater than 2.5 mm day$^{-1}$ compared to dry season (NH: NDJFM; SH: MJJAS) are shown for the LIG (red contours) and for PI (blue contours).





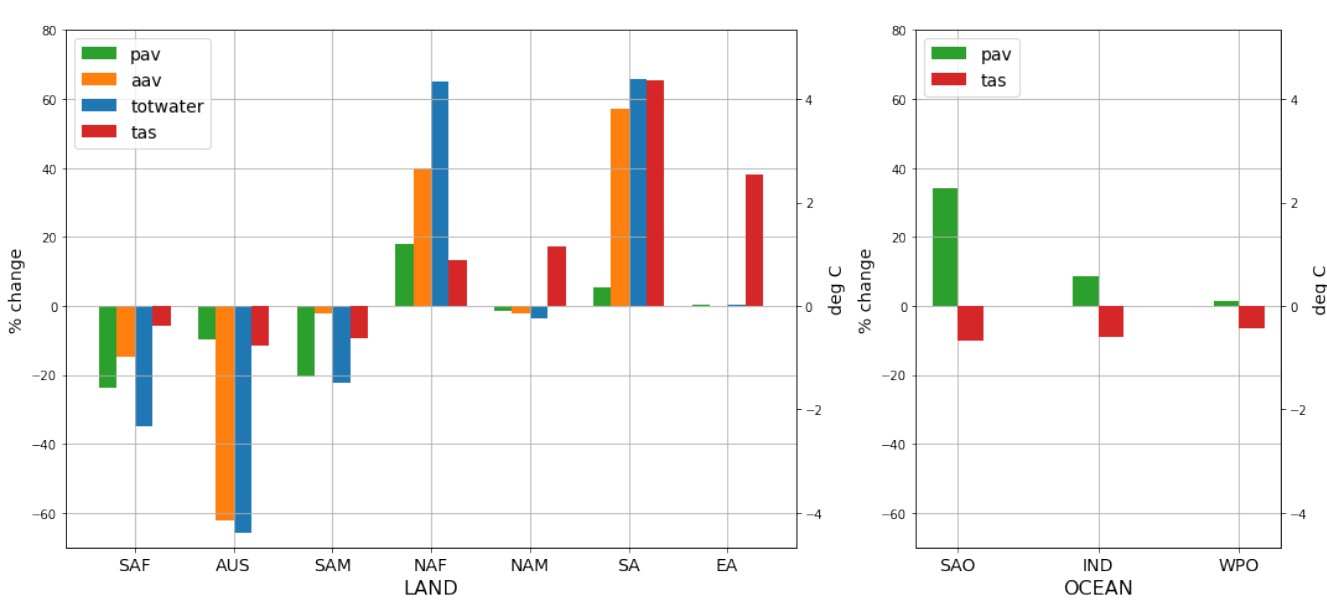

**Figure 7.** Changes in monsoon season (NH: MJJAS, SH: NDJFM) precipitation averaged within (left) terrestrial monsoon domains shown in Fig. 6 and (right) over southern hemispheric ocean regions. Areas in which individual monsoon and ocean regions are bounded are defined in Table S1. Pav: Percentage change in area-averaged precipitation rate during monsoon season. Aav: Percentage change in areal extent of regional monsoon domain. Totwater: Percentage change in total precipitated water during monsoon season (mean precipitation rate over monsoon domain multiplied by areal extent). TAS: air surface temperature (in °C). Key: SAF = South African monsoon; AUS = Australian monsoon; SAM = South American monsoon; NAF: North African monsoon; NAM = North American monsoon; SA = South Asian monsoon; EA = East Asian monsoon; SAO = South Atlantic Ocean; IND = Southern Indian Ocean; WPO = Western to Central Equatorial Pacific Ocean.



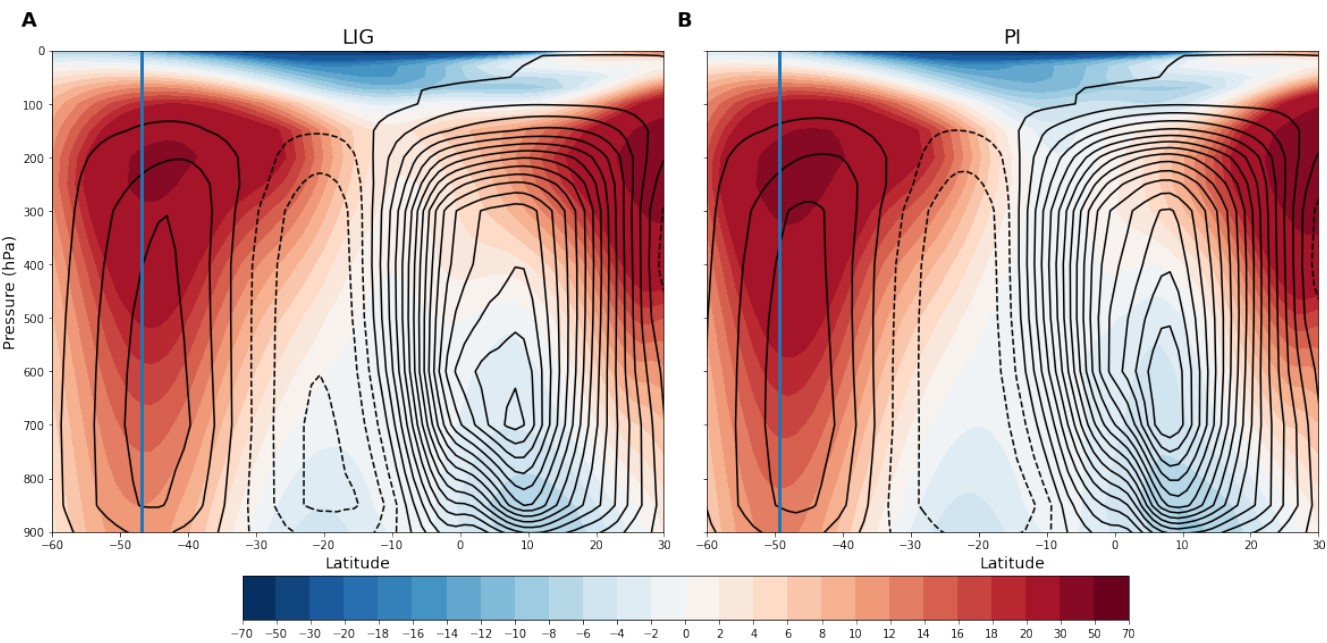

**Figure 8.** Atmospheric mass stream function in DJF, represented by black contour lines, is contoured every 30 Sv, with the zero contour omitted. Zonal mean zonal winds (m s$^{-1}$) is represented by colour shading. (a): LIG; (b): PI. Please note the shading interval is not linear. The vertical blue lines are the location of SH eddy-driven jet, defined as the maximum zonally-averaged zonal wind at 850 hPa (Ceppi et al., 2013).



**Table 1.** Forcings and boundary conditions of the piControl and lig127k experiments.

|  | piControl (1850 CE) | lig127k (127 ka) |
| --- | --- | --- |
| **Orbital parameters** |  |  |
| Eccentricity | 0.016764 | 0.039378 |
| Obliquity (degrees) | 23.459 | 24.040 |
| Perihelion - 180 | 100.33 | 275.41 |
| Vernal equinox | Fixed to noon on 21 March | Fixed to noon on 21 March |
| **Greenhouse gases** |  |  |
| Carbon dioxide (ppm) | 284.3 | 275 |
| Methane (ppb) | 808.2 | 685 |
| Nitrous oxide (ppb) | 273.0 | 255 |
| Other GHGs | CMIP DECK piControl | 0 |
| Solar constant | TSI: 1365.65 W m$^{-2}$ * | |
| Paleogeography | Modern | |
| Ice sheets | Modern | |
| Vegetation | CMIP DECK piControl | |
| Aerosols: dust, volcanic, etc. | CMIP DECK piControl | |

*While the solar constant in the protocol for CMIP DECK piControl is 1360.747 W m$^{-2}$, both the piControl and lig127k experiments were integrated with 1365.65 W m$^{-2}$ according to CMIP5-PMIP3 guidelines. This allows comparison between our piControl and our lig127k experiment.



**Table 2.** Model components of the ACCESS-ESM1.5.

| Model component | Name | Resolution |
|---|---|---|
| Atmosphere | UM7.3 | 1.875° x 1.25°, with 38 vertical levels, extending to 40 km. |
| Land surface | CABLE2.4 | Same horizontal resolution as atmosphere. Each grid cell comprises of 13 plant functional types (PFT). No dynamic vegetation, but interactive LAI. Biogeochemistry implemented by CASA-CNP module. |
| Coupler | OASIS-MCT | |
| Ocean | MOM5 | Nominally 1° in horizontal, with latitudinal refinements around the Equator (0.33° between 10° S and 10° N) and the Southern Ocean (ranging from 0.25° at 78° S to 1° at 30° S). 50 vertical levels with a nominal 10 m thickness in the upper ocean. |
| Marine carbon cycle | WOMBAT | Same as MOM5 |
| Sea ice | CICE4.1 | Same as MOM5 |