# Peer review of "Land-sea temperature contrasts at the Last Interglacial and their impact on the hydrological cycle"

_Climate of the Past, 2020_

## Referee Comment (RC1) · Josephine Brown (Referee) · 22 Dec 2020

Review of "Weak Southern Hemispheric monsoons during the Last Interglacial Period" by Nicholas K. H. Yeung et al.

This is a very interesting paper which clearly and concisely describes the results of a new Last Interglacial simulation with the ACCESS-ESM1.5 CMIP6/PMIP4 model. The model results are compared with both other CMIP6/PMIP4 simulations and proxy records. The main focus of the paper is on changes in the Southern Hemisphere precipitation in monsoons and convergence zones, although global changes are also summarised. The paper is well structured and clearly written, providing a comprehensive discussion of previous model and proxy studies. The figures are also clear and

well described. I found the discussion of the results convincing and thorough. I have only a few minor corrections to suggest, listed below.

Specific comments:

1. Title – the paper covers a wider scope than suggested by the title, perhaps the title could be modified to reflect this.

2. Line 205-206 (also Figure 6 caption): The definition of monsoon domain is a bit confusing. The meaning is that the difference between wet and dry season precipitation is greater than 2.5 mm/day but the wording implies that wet season precipitation itself is greater than 2.5 mm/day. Rewrite to clarify.

3. Line 226-228: Interesting that there is also an increase in precipitation at the south-eastern edge of the SPCZ, associated with the weakening of the south-eastern Pacific climatological high pressure region and anti-cyclonic circulation (whereas this region is projected to become drier in future climate simulations).

4. Line 249: There is also reduced north-westerly flow to north-west of Australia.

5. Line 266: How was southern boundary of the Hadley cell defined?

6. Line 301-302: Apologies for suggesting citation of my own work, but Brown et al. (2020) is relevant here in discussing proxy records of last interglacial ENSO and latest PMIP4 model simulations – that study found that a large majority of models simulate weakened ENSO in the lig127k simulations. You could also cite other proxy & model studies of last interglacial ENSO mentioned therein.

7. Line 320-321: Another study which considered the response of SH monsoons to past and future forcing is D'Agostino et al. (2020) (again, apologies as I am a co-author), who found reduced SH monsoon strength in the mid-Holocene but increases in future under RCP8.5.

8. Figure 6: See comment re line 205, make monsoon definition clearer.

9. Figure 8: Why not identify the edge of the Hadley cell instead, as this is discussed in the text but the eddy-driven jet is not.

Technical corrections:

1. Abstract, line 4: "southern hemispheric" – replace with "Southern Hemisphere" here and elsewhere.

2. Line 194: "globally averaged" should be "zonally averaged" (referring to Figure 5).

3. Line 206: replace "general monsoon season" with "wet season" or "monsoon wet season".

4. Line 209: replace "general extension" with "expansion".

5. Line 230: eastern Australia (not Eastern).

6. Line 281: replace "multi-model mean" with "model range"

7. Line 341 onwards: References: Check formatting as several have duplicate doi/URL information.

8. Figure 1: Specify that months are calendar-adjusted.

9. Figure 4: Label panels a, b, c, d etc.

References: Brown, J. R., Brierley, C. M., An, S.-I., Guarino, M.-V., Stevenson, S., Williams, C. J. R., Zhang, Q., Zhao, A., Abe-Ouchi, A., Braconnot, P., Brady, E. C., Chandan, D., D'Agostino, R., Guo, C., LeGrande, A. N., Lohmann, G., Morozova, P. A., Ohgaito, R., O'ishi, R., Otto-Bliesner, B. L., Peltier, W. R., Shi, X., Sime, L., Volodin, E. M., Zhang, Z., and Zheng, W.: Comparison of past and future simulations of ENSO in CMIP5/PMIP3 and CMIP6/PMIP4 models, Clim. Past, 16, 1777–1805, https://doi.org/10.5194/cp-16-1777-2020, 2020.

D'Agostino, R., Brown, J. R., Moise, A., Nguyen, H., Dias, P. L. S., & Jungclaus, J. (2020). Contrasting Southern Hemisphere Monsoon Response: MidHolocene Orbital

[Figure]

Forcing versus Future Greenhouse Gas–Induced Global Warming, Journal of Climate, 33(22), 9595-9613, https://doi.org/10.1175/JCLI-D-19-0672.1.

---

## Referee Comment (RC2) · Anni Zhao (Referee) · 19 Jan 2021

In this manuscript, the authors, Yeung et al., provide an analysis of weakening monsoons in the Southern Hemisphere at the Last Interglacial in the ACCESS-ESM1.5 lig127k and piControl simulations. The manuscript is well written and I would like to suggest this manuscript to be accepted with some minor revision.

My minor comments regarding the text and analysis are following:

1. Section 3: This section includes more results than monsoons in the Southern Hemisphere. Please modify, or change your title.

2. Figure 2d: Remove the red vertical lines around 100degW and 80degW in high latitudes.

[Figure]

3. Figure S1 and sea ice extent in text: The numbers of sea ice extent plotted in Figure S1 are similar to the sea ice areas (not extent) plotted in Figure 4 in Otto-Bliesner et al. (2020). I'm a co-author of that LIG paper, and we may have made an error in our calculation as our numbers are different from yours. My apologies. Following the SIMIP community, we've gone for area not extent in the LIG paper. The SIMIP papers appear to ditched "sea ice extent" (area of grid boxes with >15% coverage) in favour of "sea ice area" (grid box area * fractional coverage). This decision is motivated by sea ice area being better constrained by the observations. You might want to consider this.

4. Line 128-129: Could you give a potential explanation to the mismatch? Meltwater from ice sheets, in line 292-295, could be one.

5. The structure of section 3.2.2 is confusing. Figures and text mix up analysis of DJF precipitation and NDJFM precipitation, in my view.

6. Line 105-108 and Section 3.2.2: If I understand correctly, adjusted monthly precipitation data are used in analysing monsoon variables. Brierley et al (2020) (I'm also a co-author, my apologies) estimated the size of the interpolation error from the PaleoCalAdjust routine on monsoon domain. The calendar adjustment brings a dry bias in monsoon variables and its magnitude is larger than the wet bias in directly using monthly data. Otto-Bliesner et al (2020) and Brierley et al (2020) therefore not applied the calendar adjustment in monsoon analysis in the lig127k and midHolocene simulations. I attach our LIG monsoon numbers used in Otto-Bliesner et al (2020) that have been calculated from your simulations in the supplement. I can't say which one is correct.

7. Line 205-209: The definition of monsoon domain of Wang et al (2011) includes two parts: first, summer rainfall minus winter rainfall is at least 2.5 mm/day; second, at least 55% of the annual rainfall falls in the summer season. Could you justify why only the first part has been used?

8. Figure 6: Remove the white bands at the Greenwich Meridian.

9. Figure 7: The right-hand axes should be in red, so that it is clear that they are only associated with the tas bars.

10. You might want to refer to the description of SH monsoon in D'Agostino et al (2020) to help with the mechanisms in discussion.

Refs: Brierley, C. M., Zhao, A., Harrison, S. P., Braconnot, P., Williams, C. J. R., Thornalley, D. J. R., Shi, X., Peterschmitt, J.-Y., Ohgaito, R., Kaufman, D. S., Kageyama, M., Hargreaves, J. C., Erb, M. P., Emile-Geay, J., D'Agostino, R., Chandan, D., Carré, M., Bartlein, P. J., Zheng, W., Zhang, Z., Zhang, Q., Yang, H., Volodin, E. M., Tomas, R. A., Routson, C., Peltier, W. R., Otto-Bliesner, B., Morozova, P. A., McKay, N. P., Lohmann, G., Legrande, A. N., Guo, C., Cao, J., Brady, E., Annan, J. D., and Abe-Ouchi, A.: Large-scale features and evaluation of the PMIP4-CMIP6 midHolocene simulations, Clim. Past, 16, 1847–1872, https://doi.org/10.5194/cp-16-1847-2020, 2020.

D'Agostino, R., Brown, J. R., Moise, A., Nguyen, H., Silva Dias, P. L., & Jungclaus, J. : Contrasting Southern Hemisphere Monsoon Response: MidHolocene Orbital Forcing versus Future Greenhouse Gas–Induced Global Warming, Journal of Climate, 33(22), 9595-9613, 2020.

Please also note the supplement to this comment:
https://cp.copernicus.org/preprints/cp-2020-149/cp-2020-149-RC2-supplement.pdf

**Supplement:**

| Monsoon | pav | aav | totwater |
|---------|------|------|----------|
| NAF | 24.0003073 | 43.5295177 | 77.8094119 |
| EAS | 11.031978 | 26.7931956 | 40.8224329 |
| AUSMC | -14.530157 | -44.336891 | -53.134299 |
| NAMS | 4.82723199 | -10.790797 | -6.3183614 |
| SAF | -16.944443 | -9.7686687 | -25.057758 |
| SAMS | -11.781151 | 1.04084481 | -10.896088 |
| SAS | -19.147591 | 33.9799021 | 8.52394403 |

---

## Author Comment (AC1) · 3 Mar 2021

**Response to Reviewer 1**

Review of "Weak Southern Hemispheric monsoons during the Last Interglacial Period" by Nicholas K. H. Yeung et al.

This is a very interesting paper which clearly and concisely describes the results of a new Last Interglacial simulation with the ACCESS-ESM1.5 CMIP6/PMIP4 model. The model results are compared with both other CMIP6/PMIP4 simulations and proxy records. The main focus of the paper is on changes in the Southern Hemisphere precipitation in monsoons and convergence zones, although global changes are also summarised. The paper is well structured and clearly written, providing a comprehensive discussion of previous model and proxy studies. The figures are also clear and well described. I found the discussion of the results convincing and thorough. I have only a few minor corrections to suggest, listed below.

We would like to thank the Referee, Dr Josephine Brown, for her positive and constructive comments, which helped improve the manuscript. Responses to each individual comment are included below.

Specific comments:
1. Title – the paper covers a wider scope than suggested by the title, perhaps the title could be modified to reflect this.
We would like to thank Dr Brown for this suggestion. We have changed the title to: "*Land-sea temperature contrasts at the Last Interglacial and their impact on the hydrological cycle*"

2. Line 205-206 (also Figure 6 caption): The definition of monsoon domain is a bit confusing. The meaning is that the difference between wet and dry season precipitation is greater than 2.5 mm/day but the wording implies that wet season precipitation itself is greater than 2.5 mm/day. Rewrite to clarify.
The definition has been clarified. As reviewer #2 suggested including a criterion that at least 55% of annual precipitation occurs during wet season, the sentence has been rewritten as "*Monsoon domains are defined as regions in which the precipitation during the monsoon season is (1) greater than the dry season by at least 2.5 mm/day, and (2) responsible for at least 55% of the annual precipitation (Wang et al., 2011).*" The caption of Figure 6 has been modified to reflect this revised definition of monsoon domains.

3. Line 226-228: Interesting that there is also an increase in precipitation at the southeastern edge of the SPCZ, associated with the weakening of the south-eastern Pacific climatological high pressure region and anti-cyclonic circulation (whereas this region is projected to become drier in future climate simulations).
The following has been added to Section 3.2.2: "Precipitation also increases at the southeastern edge of the SPCZ (Figs. 4c, 6b). This is linked to the weakening of the south Pacific high (Fig. 4f) and the associated westerlies south of 40°S and the easterlies between 15°S and 35°S (Fig. 6b)."

Also, the following has been included in the discussion section: "Incidentally, lig127k shows increased precipitation at the southeastern edge of the SPCZ, which is associated with the weakening of the local high pressure region. Such enhanced moisture at the eastern SPCZ margin is also linked to a reduction in trade wind inflow from the southeastern Pacific (Fig. 6b), as previously demonstrated by Lintner and Neelin (2008). Interestingly, CMIP5 models project drier conditions over the southeastern edge of the SPCZ due to the transport of dry subtropical air into the region (Brown et al., 2020b), thus suggesting a different mechanism can take place in a warmer climate."

4. Line 249: There is also reduced north-westerly flow to north-west of Australia.
This reduction is now being included in the sentence: "Due to reduced land-sea temperature gradients, colder conditions over land induce a weakening of the onshore winds (i.e. a weakening of the easterlies over Brazil, South Africa, and northeastern Australia, and reduced north-westerlies in northwestern Australia), which tends to decrease moisture advection inland, and restrict convective activity over the SH land regions."

5. Line 266: How was southern boundary of the Hadley cell defined?

The southern boundary of the Hadley cell was estimated as the zero contour line between 30 and 40S (Fig. R1 in this document). The zero contours of the mass stream function have been now included in Figure 8 for easier visualisation.

[Figure]

*Figure R1: Atmospheric mass stream function in DJF, represented by solid (positive) and dashed (negative) contour lines, is contoured every 20 Sv, with the bold lines being the zero contours. Zonal mean zonal winds (m s−1) are represented by colour shading. (a): LIG; (b): PI. Please note the shading interval is not linear.*

6. Line 301-302: Apologies for suggesting citation of my own work, but Brown et al. (2020) is relevant here in discussing proxy records of last interglacial ENSO and latest PMIP4 model simulations – that study found that a large majority of models simulate weakened ENSO in the lig127k simulations. You could also cite other proxy & model studies of last interglacial ENSO mentioned therein.
We agree that Brown et al. (2020) is relevant concerning the discussion of ENSO in the Last Interglacial. A sentence has been added in this paragraph: "ENSO variability was shown to be consistently reduced during the LIG compared to PI, with the ensemble mean of PMIP4 simulations suggesting a 20% weakening of ENSO amplitude (Brown et al., 2020)."

7. Line 320-321: Another study which considered the response of SH monsoons to past and future forcing is D'Agostino et al. (2020) (again, apologies as I am a coauthor), who found reduced SH monsoon strength in the mid-Holocene but increases in future under RCP8.5.
A sentence citing D'Agostino et al. (2020) is added: "In fact, moisture budget analysis has shown that SH monsoon expands and intensifies under the RCP8.5 scenario (D'Agostino et al., 2020)."

A paragraph discussing the results of D'Agostino et al. (2020) is also added in the Discussion section, as suggested by Reviewer #2:
"The orbital forcings and latitude-month insolation anomalies relative to PI are similar between the LIG and mid-Holocene (6 ka). SH monsoons are also weakened and contracted during the mid-Holocene (D'Agostino et al., 2020), with similar SH land-sea precipitation contrast patterns compared to lig127k. The reduced monsoonal precipitation in the mid-Holocene is largely attributed to changes in atmospheric mean flow (i.e. the Walker-Hadley circulation) and the decrease in net energy input (D'Agostino et al., 2020). While an energetic approach is beyond the scope of this study, the weakening of SH monsoons in our lig127k simulation are indeed associated with weaker local insolation during the wet season, which leads to reduced surface air temperature (Fig. 2a) and increased surface pressure over land (Fig. 4f). One can therefore hypothesise that the weakening of SH monsoons in lig127k might also be associated with a similar decrease in net energy input. Furthermore, as the changes in the Walker-Hadley circulation are not analysed in detail here, a closer examination of the relationship between SH monsoons and atmospheric circulation, such as the Hadley circulation across individual SH monsoon domains, is recommended for future studies."

8. Figure 6: See comment re line 205, make monsoon definition clearer.
The definition has been clarified (see response to comment #2).

9. Figure 8: Why not identify the edge of the Hadley cell instead, as this is discussed in the text but the eddy-driven jet is not.
The vertical line marking the eddy-driven jet is now removed. Bold lines marking the zero contours are used to identify the edge of the Hadley cells.

Technical corrections:

All the technical corrections below have been applied in the manuscript.

1. Abstract, line 4: "southern hemispheric" – replace with "Southern Hemisphere" here and elsewhere.
2. Line 194: "globally averaged" should be "zonally averaged" (referring to Figure 5).
3. Line 206: replace "general monsoon season" with "wet season" or "monsoon wet season".
4. Line 209: replace "general extension" with "expansion".
5. Line 230: eastern Australia (not Eastern).
6. Line 281: replace "multi-model mean" with "model range"
7. Line 341 onwards: References: Check formatting as several have duplicate doi/URL information.
8. Figure 1: Specify that months are calendar-adjusted.
9. Figure 4: Label panels a, b, c, d etc.

---

## Author Comment (AC2) · 3 Mar 2021

**Response to Reviewer 2**

In this manuscript, the authors, Yeung et al., provide an analysis of weakening monsoons in the Southern Hemisphere at the Last Interglacial in the ACCESS-ESM1.5 lig127k and piControl simulations. The manuscript is well written and I would like to suggest this manuscript to be accepted with some minor revision.

We would like to thank the Referee, Dr Anni Zhao, for her positive and constructive comments, which helped improve the manuscript. Responses to each individual comment are included below.

My minor comments regarding the text and analysis are following:

1. Section 3: This section includes more results than monsoons in the Southern Hemisphere. Please modify, or change your title.
We would like to thank Dr Zhao for this suggestion. We have changed the title to: "*Land-sea temperature contrasts at the Last interglacial and their impact on the hydrological cycle*"

2. Figure 2d: Remove the red vertical lines around 100degW and 80degW in high latitudes.
The lines are now removed.

3. Figure S1 and sea ice extent in text: The numbers of sea ice extent plotted in Figure S1 are similar to the sea ice areas (not extent) plotted in Figure 4 in Otto-Bliesner et al. (2020). I'm a co-author of that LIG paper, and we may have made an error in our calculation as our numbers are different from yours. My apologies. Following the SIMIP community, we've gone for area not extent in the LIG paper. The SIMIP papers appear to ditched "sea ice extent" (area of grid boxes with >15% coverage) in favour of "sea ice area" (grid box area * fractional coverage). This decision is motivated by sea ice area being better constrained by the observations. You might want to consider this.
Figure S1 was showing sea ice area indeed, instead of extent. A mistake was made when producing the plot title, and the word "extent" was mistakenly used in the text. The numbers are therefore in line with Otto-Bliesner et al. (2020). We apologise for the mistake and they have been renamed as "sea-ice area".

4. Line 128-129: Could you give a potential explanation to the mismatch? Meltwater from ice sheets, in line 292-295, could be one.
A potential explanation is now included: "Paleoproxy records suggest there was no deep-water formation in the Labrador Sea during the LIG (Hillaire-Marcel et al., 2001). There is also evidence of meltwater discharge from the Greenland ice-sheet during the early part of the LIG (Galaasen et al., 2014 and 2020, Tzedakis et al., 2018), which could have suppressed deep ocean convection in the Labrador Sea and led to a cooling there (e.g. Tzedakis et al., 2018), thus explaining the discrepancy."

5. The structure of section 3.2.2 is confusing. Figures and text mix up analysis of DJF precipitation and NDJFM precipitation, in my view.

Thank you for your comment. To avoid confusion, the discussion of monsoon activities in Section 3.2.2 is now based on the precipitation results during NDJFM and MJJAS, and Figure 6 has been changed accordingly. We find the now amended precipitation results (NDJFM/MJJAS) in Figure 6 are very similar to the original (DJF/JJA), with the magnitude of the anomalies only slightly lowered. As such, this does not affect our analysis and conclusions.

6. Line 105-108 and Section 3.2.2: If I understand correctly, adjusted monthly precipitation data are used in analysing monsoon variables. Brierley et al (2020) (I'm also a co-author, my apologies) estimated the size of the interpolation error from the PaleoCalAdjust routine on monsoon domain. The calendar adjustment brings a dry bias in monsoon variables and its magnitude is larger than the wet bias in directly using monthly data. Otto-Bliesner et al (2020) and Brierley et al (2020) therefore not applied the calendar adjustment in monsoon analysis in the lig127k and midHolocene simulations. I attach our LIG monsoon numbers used in Otto-Bliesner et al (2020) that have been calculated from your simulations in the supplement. I can't say which one is correct.

Thank you for raising this point. The figure below (Figure R2 in this document) is similar to Figure 7 in the manuscript but with non-calendar-adjusted results. Compared to using calendar-adjusted results,

changes with respect to piControl in all precipitation-related parameters (pav, aav, totwater) are largely similar. For SH monsoons, calendar-adjusted results are slightly dry-biased (less than 5% decrease compared to non-calendar-adjusted). For NH monsoons, differences are even smaller, except for the decrease in NAF total-water-precipitated (decreases from ~90% to ~72%).

Since we do not have daily data and the results are shown to be largely similar, we decided to keep the calendar-adjusted results in the main text, but have included an alternate version of Figure 7 using non-calendar-adjusted results in the Supplementary Material (now Figure S3). The following has been added to Section 3.2.2: "The metrics on Fig. 7 have been calculated using an adjusted paleo-calendar from monthly outputs (Bartlein and Shafer, 2019). However, Brierley et al. (2020) suggest that this method might accentuate the anomalies. As seen in Fig. S3, non-calendar adjusted results are similar to the calendar adjusted ones (differences ~<5 %), even though the calendar-adjusted results are slightly dry-biased for most precipitation-related metrics. The major difference is the total precipitated water for the North African monsoon (NAF), in which the calendar adjusted suggest a ~90% increase, whereas the non-calendar adjusted suggest a ~72% increase."

[Figure]

*Figure R1: Figure 7 in manuscript, but with non-calendar adjusted data.*

7. Line 205-209: The definition of monsoon domain of Wang et al (2011) includes two parts: first, summer rainfall minus winter rainfall is at least 2.5 mm/day; second, at least 55% of the annual rainfall falls in the summer season. Could you justify why only the first part has been used?
Thank you for your suggestion. The 55% definition is now included in our analysis: "Monsoon domains are defined as regions in which the precipitation in monsoon season is (1) greater than dry season by at least 2.5 mm/day, and (2) responsible for at least 55% of annual precipitation (Wang et al., 2011)." Resulting changes are shown to be minor (see amended Figure 6 and 7) and do not affect our conclusions.

8. Figure 6: Remove the white bands at the Greenwich Meridian.
The bands are now removed.

9. Figure 7: The right-hand axes should be in red, so that it is clear that they are only associated with the tas bars.
The tick labels in the right-hand axes are now in red.

10. You might want to refer to the description of SH monsoon in D'Agostino et al (2020) to help with the mechanisms in discussion.
The following paragraph has been added to the discussion:
"The orbital forcings and latitude-month insolation anomalies relative to PI are similar between the LIG and mid-Holocene (6 ka). SH monsoons are also weakened and contracted during the mid-Holocene (D'Agostino et al., 2020), with similar SH land-sea precipitation contrast patterns compared to lig127k. The reduced monsoonal precipitation in the mid-Holocene is largely attributed to changes in atmospheric mean flow (i.e. the Walker-Hadley circulation) and the decrease in net energy input (D'Agostino et al., 2020). While an energetic approach is beyond the scope of this study, the

weakening of SH monsoons in our lig127k simulation are indeed associated with weaker local insolation during the wet season, which leads to reduced surface air temperature (Fig. 2a) and increased surface pressure over land (Fig. 4f). One can therefore hypothesise that the weakening of SH monsoons in lig127k might also be associated with a similar decrease in net energy input. Furthermore, as the changes in the Walker-Hadley circulation are not analysed in detail here, a closer examination of the relationship between SH monsoons and atmospheric circulation, such as the Hadley circulation across individual SH monsoon domains, is recommended for future studies."